# Roles of inter- and intramolecular tryptophan interactions in membrane-active proteins revealed by racemic protein crystallography

Alexander J. Lander [1], Laura Domínguez Mercado[2], Xuefei Li[1], Irshad Maajid Taily[1], Brandon L. Findlay [2✉], Yi Jin [3✉] & Louis Y. P. Luk [1✉]

Tryptophan is frequently found on the surface of membrane-associated proteins that interact with the lipid membrane. However, because of their multifaceted interactions, it is difficult to pinpoint the structure-activity relationship of each tryptophan residue. Here, we describe the use of racemic protein crystallography to probe dedicated tryptophan interactions of a model tryptophan-rich bacteriocin aureocin A53 (AucA) by inclusion and/or exclusion of potential ligands. In the presence of tetrahedral anions that are isosteric to the head group of phospholipids, distinct tryptophan H-bond networks were revealed. H-bond donation by W40 was critical for antibacterial activity, as its substitution by 1-methyltryptophan resulted in substantial loss of activity against bacterial clinical isolates. Meanwhile, exclusion of tetrahedral ions revealed that W3 partakes in formation of a dimeric interface, thus suggesting that AucA is dimeric in solution and dissociated to interact with the phosphate head group in the presence of the lipid membrane. Based on these findings, we could predict the tryptophan residue responsible for activity as well as the oligomeric state of a distant homologue lacticin Q (48%).

[1] School of Chemistry, Cardiff University, Main Building, Park Place, Cardiff CF10 3AT, UK. [2] Department of Chemistry & Biochemistry, Richard J. Renaud Science Complex, Concordia University, Montréal, Québec H4B 1R6, Canada. [3] Manchester Institute of Biotechnology, University of Manchester, Manchester M1 7DN, UK. ✉email: brandon.findlay@concordia.ca; yi.jin@manchester.ac.uk; lukly@cardiff.ac.uk

Tryptophan is abundantly found on the surface of bacteriocins, venoms and other membrane-active proteins of medical relevance[1]. While many are highly conserved, it is often difficult to elucidate their function based only on site-directed mutagenesis. Indeed, the heterocyclic ring of tryptophan exhibits a wide spectrum of polarity that has been demonstrated to interact with various components of the lipid membrane domains[2]. It can associate with a cationic group such as choline through π-based interactions[3,4], the hydrophobic alkyl chain of a fatty acid or amino acid[5–7], or a H-bond acceptor such as the phosphate head group by donating the indole's N-H[8,9]. In line with the dynamical nature of proteins[10], a tryptophan residue may have multiple interacting partners that interchange during the course of action[2]. Indeed, while each tryptophan residue found on the surface of a protein likely plays a defined role in the structure-activity relationship, their diverse modes of interactions complicate mechanistic elucidation. A potentially straightforward approach to probe key tryptophan interactions is racemic protein crystallography[11,12]. When proteins of natural L-chirality are introduced with their synthetic D-enantiomers, they can form centrosymmetric crystals in achiral space groups with high dimensionality, thereby enabling structures solved at a high resolution over wide-ranging conditions[11–21]. Consequently, by analyzing racemic crystals of membrane-active proteins formed in the presence and/or absence of potential ligands, dedicated intra- and intermolecular interactions of tryptophan can be revealed. To prove this concept, we have conducted mechanistic analysis of a model bacteriocin aureocin A53 (AucA) through racemic protein crystallography.

Produced by its hosts as means of eliminating competitors[22,23], AucA exhibits broad-spectrum activity against Gram-positive pathogens and contains conserved tryptophan residues (W22, W31 and W40) which are found in many other homologous bacteriocins[24–26]. Previously, NMR analysis of recombinant AucA revealed a small globular morphology, in which these tryptophan residues as well as the non-conserved W3 are exposed on the protein surface[27]. While the conformationally-averaged NMR structure has not gained direct evidence for their involvement, these tryptophan have been suggested to play roles in interacting with membrane lipids.

Through outlining a cost-effective synthetic scheme of the enantiomeric AucA, here we have conducted racemic protein crystallography analysis. In the presence of tetrahedral anions (sulfate and glycerol phosphate), which are isosteric to the head group of the membrane phospholipid[28], the resulting crystal structure revealed a unique salt network whereby the oxygen atoms of the salts are H-bonded by W3, W31 and W40 and their respective neighboring lysine residues (K25, K27 and K44). Replacement of these tryptophan residues affected AucA's activity against bacterial strains isolated from clinical samples. In particular, H-bond donation by W40 was exceptionally crucial; substitution by 1-methyl tryptophan nearly abolished all AucA's activity. In the absence of the tetrahedral anion, W3 swaps to interacting with residues from another AucA forming a dimer interface. This implies that, coordinated by W3, AucA is dimeric in solution and dissociated to interact with the phosphate head group in the presence of the lipid membrane. Based on these findings, we could accurately predict the tryptophan residue responsible for activity in a distant homolog lacticin Q (LnqQ; 48% homology) as well as its oligomeric state.

## Results and discussion

**Synthesis of bacteriocin enantiomers reveals non-stereospecific antibacterial activity.** A linear solid-phase peptide synthesis (SPPS) of AucA requires a large excess of amino acids (10–20 equivalents, see Supplementary Information and ref. [29]) and is economically unsuitable for preparing its enantiomers. Therefore, total chemical synthesis involving native chemical ligation was developed (Fig. 1). In situ hydrazide activation for peptide bond formation[30,31], followed by thiol removal and desulfurization[32], yielded the bacteriocins in a one-pot procedure. Of note, preparation of the corresponding peptide segments only requires two equivalents of amino acids in each coupling step. Also, while an Ala11Cys substitution allows for both ligation and desulfurization resulting in native AucA, removal of thiol groups at other Ala positions was found to be unsuccessful, likely because the reagent could not access the targeted positions (Supplementary Fig. S1). The ligation approach could generate the enantiomeric AucA entirely composed of D-amino acids (D-AucA) which yielded an opposite optical rotation to the native L-enantiomers in circular dichroism (CD) spectroscopic analysis (Supplementary Fig. S2).

The stability of L- and D-AucA was evaluated by subjecting them to different proteases (Supplementary Fig. S3). L-AucA was found to be proteolytically resistant when treated with proteinase K. Chymotrypsin only caused degradation of L-AucA when the enzyme:substrate ratio increased from 1:20 to 10:1. Similarly, trypsin degraded L-AucA at an enzyme:substrate ratio of 10:1. In contrast, the enantiomeric D-AucA remained intact under all conditions tested (Supplementary Fig. S3). These findings suggest that the unusual chirality of D-AucA confers an even more superior resistance to proteolysis compared to its relatively stable L-counterpart.

Antibacterial activity of the enantiomeric D-AucA was assessed by determining minimum inhibitory concentrations (MICs) against a panel of bacterial pathogens isolated from hospitalized patients (CANWARD[33]; Fig. 2 & Supplementary Table S1). D-AucA is at least as active as its L-enantiomer and does not discriminate between antibiotic-sensitive and -resistant strains, suggesting that the bacteriocins target the bacterial membrane through non-stereospecific interactions[34]. In the case of S. aureus CW115852, D-AucA displayed an eight-fold increase in activity over that of its native L-enantiomer. A plausible explanation is

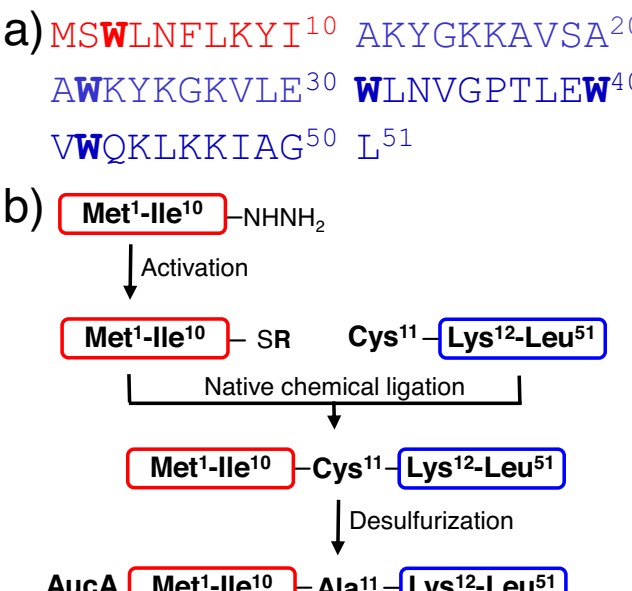

**a)** MS**W**LNFLKYI[10] AKYGKKAVSA[20]
A**W**KYKGKVLE[30] **W**LNVGPTLE**W**[40]
V**W**QKLKKIAG[50] L[51]

**b)** [Met¹-Ile¹⁰]—NHNH₂
↓ Activation
[Met¹-Ile¹⁰]— SR    Cys¹¹—[Lys¹²-Leu⁵¹]
↓ Native chemical ligation
[Met¹-Ile¹⁰]—Cys¹¹—[Lys¹²-Leu⁵¹]
↓ Desulfurization
AucA  [Met¹-Ile¹⁰]—Ala¹¹—[Lys¹²-Leu⁵¹]

**Fig. 1 Overview of the synthetic route to AucA. a** Amino acid sequence of AucA showing the contents of the peptide hydrazide (red) and N-terminal cysteine peptide (blue) segments, with tryptophan residues highlighted in bold. **b** schematic overview of the synthetic route to AucA by native chemical ligation and desulfurization.

| Antibiotic agent | | S. aureus | | | | S. epidermidis | E. faecalis |
| --- | --- | --- | --- | --- | --- | --- | --- |
| | | AC29213[b] | CW114125[c] | CW115852[c] | CW113379[c] | CW131612[c] | CW133003[c] |
| AucA variants | L-AucA | 4 | 4 | 16 | 4 | 4 | 4 |
| | D-AucA | 4 | 4 | 2 | 4 | 4 | 4 |
| | L-W3L | 4 | 4 | 32 | 4 | 4 | 16 |
| | L-¹ᴹᵉW3 | 4 | 4 | 8 | 4 | 8 | 2 |
| | L-W3E | 16 | 32 | 64 | 16 | 16 | 64 |
| | L-W22L | 2 | 8 | 16 | 16 | 4 | 16 |
| | L-W31L | 4 | 4 | 8 | 4 | 4 | 8 |
| | L-¹ᴹᵉW31 | 4 | 4 | 8 | 8 | 4 | 2 |
| | L-W31E | >64 | >64 | >64 | >64 | >64 | 32 |
| | L-W40L | >64 | >64 | >64 | >64 | >64 | 32 |
| | L-¹ᴹᵉW40 | 16 | 32 | 32 | 32 | 16 | 4 |
| | L-W40E | >64 | >64 | >64 | >64 | >64 | >64 |
| | L-W42L | 2 | 4 | 8 | 8 | 4 | 16 |
| LnqQ variants | L-LnqQ | 16 | 32 | 64 | 32 | 8 | 2 |
| | D-LnqQ | 4 | 8 | 4 | 4 | 4 | 2 |
| | L-W32L | 64 | 64 | 32 | 32 | 16 | 4 |
| | L-W41L | >64 | >64 | >64 | >64 | >64 | 64 |
| Tetracycline | | <0.125 | 32 | 8 | 16 | 1 | 1 |
| Ampicillin | | 2 | 16 | 4 | 32 | >64 | 2 |
| Melittin | | 8 | 8 | 4 | 4 | 4 | 16 |

Most active ↑ Minimum inhibitory concentration ↓ Inactive

**Fig. 2 Minimum inhibitory concentrations[a] (MIC) of AucA, LnqQ, and their variants against clinically isolated bacterial strains.** Minimum inhibitory concentrations, determined by broth microdilution, are expressed as an average of three repeat assays. Control antibiotics tetracycline, ampicillin and melittin were included in each assay. Data are color coded by concentration for clarity, with blue-green being most active (lowest MIC) and red representing loss of activity (no inhibition within assay range). [a] MIC values expressed in µg/mL, [b] AC: American Type Culture Collection strain, [c] CW: CANWARD collection clinical isolate strain.33.

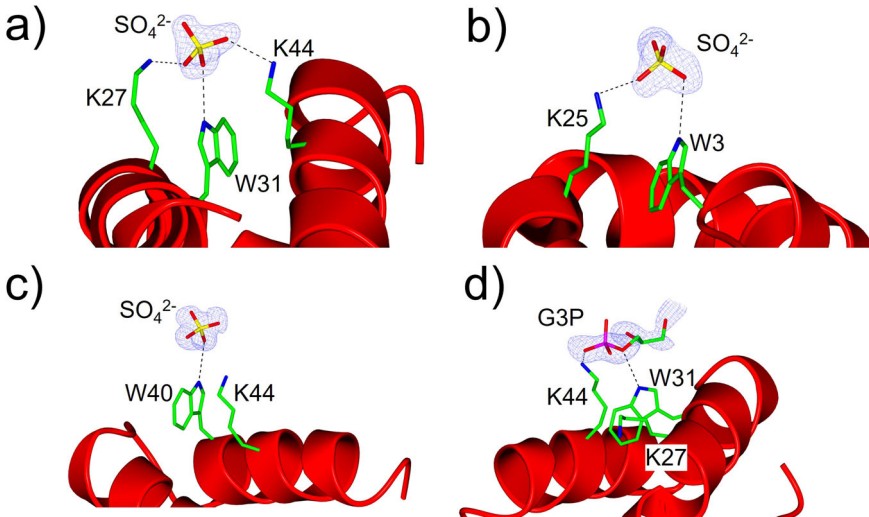

**Fig. 3 Insights from racemic protein crystallography of AucA. a–c** coordination of solvent-exposed W31, W3 and W40 with sulfate (PDB: 8AVR), **d** coordination of W31 with glycerol 3-phosphate (G3P) (PDB: 8AVT).

that the D-counterparts are more resistant to degradation. Indeed, proteases are known to be secreted by *Staphylococci* as resistance mechanisms against antibacterial peptides (see below for further analysis)[35,36].

**Tryptophan H-bonding networks revealed by racemic AucA crystallography.** Racemic protein crystallography of AucA was conducted and structures were solved with resolutions in between 0.89 and 1.21 Å. For the structures solved in the presence of sulfate, surface-exposed tryptophan residues were found to form H-bond networks on the surface of AucA. The highly conserved W31 and W40 (Supplementary Table S2) are hydrogen bonded (<3.2 Å) to the negatively charged sulfate ion along with the ammonium groups of neighboring lysine residues (K27, K44; Fig. 3a–c, PDB: 8AVR). While there are no observable anion-π or cation-π interactions between tryptophan and the sulfate or lysine side chains, the K44 aliphatic side chain associated with W40 via

van der Waals interactions, as also suggested in previous NMR studies[27], which could aid correct orientation for cooperative interaction with anions[37]. However, in contrast to the NMR analysis, the other conserved residue W22 was previously described as solvent-exposed and intramolecularly associate with the non-conserved W3[27]. In this study, this residue was found to be responsible for forming a dimeric interface (see below), and W3 was also found to associate with sulfate by coordinating with K25 (Fig. 3b, PDB: 8AVR). Indeed, when AucA crystals were grown in conditions using a high concentration of non-tetrahedral oxyanions, citrate or acetate, as replacements for the sulfate ions, the equivalent network was not observed (PDB: 8AVS). Nevertheless, when these racemic AucA crystals devoid of the Trp-anion H-bond networks were soaked with glycerol 3-phosphate, the equivalent network was observed with the W31-K44 pair (Fig. 3d, PDB: 8AVT). Since sulfate and glycerol 3-phosphate are isosteric to many of the phospholipid head

groups in bacterial membranes, these observations suggest a physiological role of W3, W31 and W40 that interact with the membrane lipid through H-bond donation."

Substitutions of the tryptophan residues were conducted to investigate the importance of the H-bond network revealed from the racemic protein crystallography analysis. The initial assessment was conducted by individually replacing each tryptophan residue in AucA with leucine, which retains the hydrophobic properties but eliminates potential H-bond and π-based interactions. Characterizations by LC-MS confirmed the synthesis of the variants, and CD spectroscopic analyses indicated that their alpha-helical folds are retained (Supplementary Fig. S4). The MIC assay indicated that the replacement of the W40 with leucine nearly abolished AucA's antibacterial activity, highlighting the importance of the indole motif at this position (Fig. 2 and Supplementary Table S1). Conversely, the AucA-W31L variant remains nearly as active as that of the wild type. A plausible explanation is that lipid interaction by W31 is subsumed by a neighboring residue. Alternatively, the hydrophobic isobutyl group in the W31L substitution could fulfill the role of lipid interaction. Trp3, though not conserved (Supplementary Table S2), was also found to form the H-bond networks with tetrahedral anion. Interestingly, the W3L variant appeared to affect AucA activity in a strain-dependent manner, showing a similar trend to that of W22L (see below for further investigation). W42 locates at the protein interior and its replacement with leucine did not affect activity.

To further investigate the interaction of W3, W31 and W40 with tetrahedral anions, they were substituted with glutamate. The carboxylate sidechain was expected to form ionic pairs with the lysine residues, resulting in salt bridges and mitigating intermolecular interaction with a tetrahedral anion[38]. Replacement of the highly conserved W40 and W31 with glutamate abolished nearly all of the antimicrobial activity (Fig. 1 and Supplementary Table S1). In turn, glutamate replacement of the non-conserved W3 hampered the antibacterial activity to a lesser extent (Supplementary Table S2). While CD spectroscopic analysis indicated that these variants have retained their secondary structure (Supplementary Fig. S4), the glutamate substitution may electrostatically disturb the microenvironment of AucA. Accordingly, these tryptophan residues were also replaced with the 1-methyl tryptophan [1Me]W, which retains the indole aromaticity but lacks the ability to interact with H-bond acceptors (See Supplementary Information for their preparation and characterization). When this unnatural amino acid was inserted into the surface exposed W40, there was a substantial loss of activity, pinpointing H-bond donation by indole at this position as critical for antibacterial activity. W40 is thus likely responsible for AucA association with the head group of phospholipids. Analogous to the leucine replacement above, methylation at positions W3 and W31 showed minimal effect on activity.

**Tryptophan interactions that are responsible for oligomerization and proteolytic resistance.** The crystal structure obtained in the absence of tetrahedral ions also revealed insights into the interactions involving W3 and W22. The protein appeared to be homodimeric with two L-AucA molecules (or two D-AucA molecules) arranged in a head-to-tail fashion (Fig. 4a, PDB: 8AVU). Unlike the intramolecular W3-W22 association reported in previous studies[27], in this dimeric configuration the indole ring of W22 shifts orientation to partake intermolecular hydrophobic interaction with L4' and L7' in the adjacent AucA molecule (Fig. 4b and Supplementary Fig. S5). Indeed, examination of the structure in protein interfaces, surfaces and assemblies[39] software

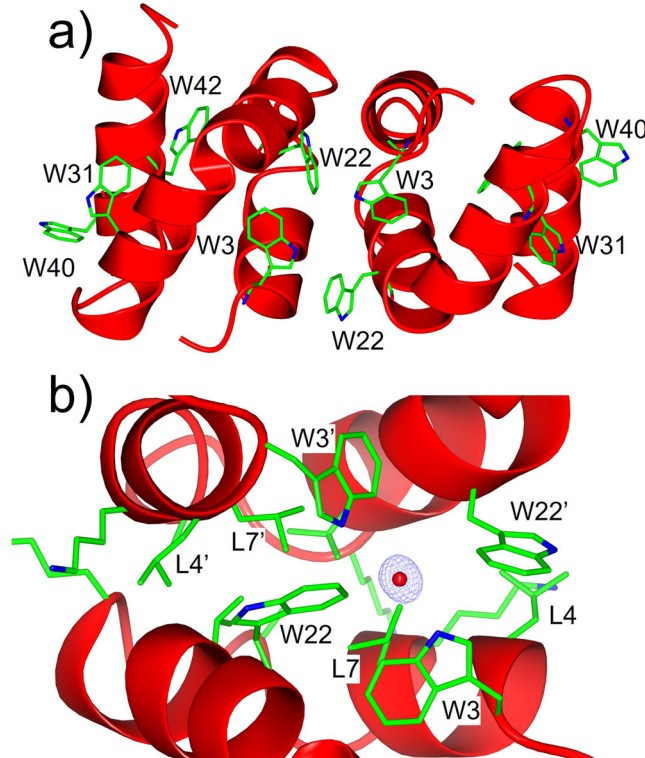

**Fig. 4 Dimeric AucA configuration revealed by racemic protein crystallography. a** Orientation of AucA molecules in the homodimeric configuration, showing location of Trp residues. **b** dimeric interface is driven by hydrophobic interactions (W22, L4', L7') and intermolecular H-bond network between adjacent W3 residues, showing bridging water molecule (red) (PDB: 8AVU). σA-weighted $2F_o$-$F_c$ omit electron density map for the bridging water is shown in blue and contoured at 1σ (0.46 electrons per Å$^3$).

showed that the dimeric interface is driven by the shielding of a hydrophobic patch on the surface of AucA from the solvent (Supplementary Fig. S6). Meanwhile, W3 forms an intermolecular H-bond network with its adjacent W3' bridged by a water molecule (Fig. 3b, PDB: 8AVU). In the presence of sulfate, this dimeric interface was replaced by the sulfate salt interaction with W3 and K25, as described above. Accordingly, W3 may oscillates its interactions between a water molecule and the head group of phospholipids.

To assess the oligomeric state in solution, the AucA-W3L variant was subjected to size exclusion analysis in phosphate-buffered saline (pH 7.4). Its retention time was clearly longer than those of the wild-type AucA (Fig. 5) and other W → L/[1Me]Trp variants which are evaluated to be dimeric (Supplementary Fig. S7), indicating that W3 plays an important role in the oligomerization. The AucA-W3[1Me]W variant appeared to exchange in between the monomeric and dimeric forms in solution. It is likely that the methyl groups can partially occupy the void of the bridging water molecule, leading to only partial disruption of the dimeric interface. W22 is the other surface-exposed tryptophan found in the dimeric interface. Its replacement by a leucine residue did not affect the retention time in the size exclusion chromatography, which supports its proposed role in associating L4' and L7' via hydrophobic interactions. Indeed, in the dimeric interface, W22 is considerably more buried from the solvent than W3, with buried surface areas of 86 Å$^2$ and 32 Å$^2$, respectively (Supplementary Table S3). Interestingly, the AucA-W22L variant is considerably more susceptible to proteolysis than the wild-type AucA, being fully digested within 24 h at physiological conditions (37 °C and pH 7.4) by proteinase K, a

promiscuous protease that recognizes both tryptophan and leucine for hydrolysis (Fig. 6 and Supplementary Fig. S8). Accordingly, these experiments suggest W3 and W22 roles in oligomerization and proteolytic resistance respectively, which likely affect AucA's ability in delivering its activity within biological contexts (bacterial cell surface). Hence, their replacement affected AucA's activity in a strain-dependent manner as observed in the MIC assay (Fig. 2 and Supplementary Table S1).

**Validation of the tryptophan interactions in a distant protein homolog.** Testing if the insights obtained from the study of AucA can be transferred to improve our understanding of other bacteriocins, we prepared a distant homolog, lacticin Q (LnqQ), for analysis (48% homology, Supplementary Table S2). A synthetic scheme based on chemical ligation was developed and the structure of LnqQ was solved by racemic protein crystallography (0.96 Å). In agreement from previous work[27], the structure of LnqQ is homologous to AucA (Supplementary Fig. S2 for CD spectra, PDB 7P5R) with multiple tryptophan residues found on the protein surface including W21, W23, W32 and W41. However, LnqQ lacks the corresponding W3 residue (Supplementary Table S2) and, as anticipated, was found to be monomeric: sharing essentially the same retention time as that of AucA-W3L in the SEC analysis (Fig. 4). Indeed, the monomeric L-LnqQ

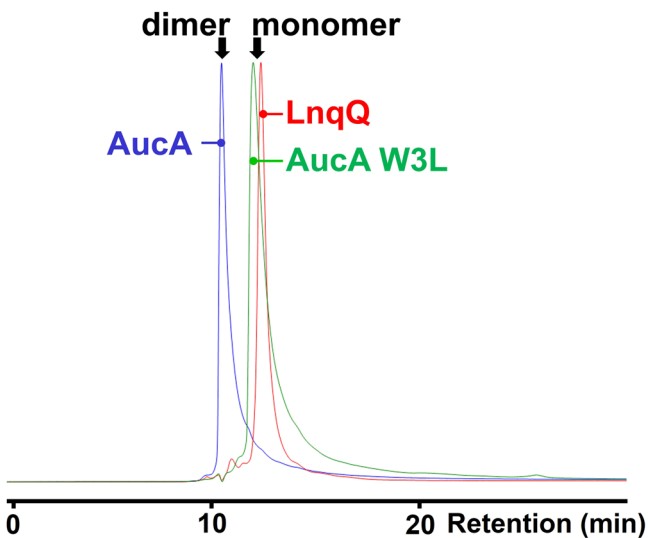

**Fig. 5 Analytical size exclusion chromatography of AucA-wt, Auc-W3L and LnqQ.** Size exclusion chromatography traces (UV 210 nm) of AucA (blue), AucA-W3L (green) and LnqQ (red), highlighting the increased retention of the AucA-W3L variant corresponding to a smaller hydrodynamic radius (monomeric state).

appeared to be more sensitive to proteolytic degradation (Supplementary Fig. S3), as its enantiomeric D-counterpart is resistant to proteolytic degradation and is significantly more potent leading up to 16-fold of difference in activity (Fig. 2 and Supplementary Table S1). Furthermore, W41, which is equivalent to W40 in AucA (Supplementary Table S2), is anticipated to be most crucial for activity, and its replacement by a leucine residue nearly demolished all LnqQ's antibacterial activity likely due to a lack of H-bond donation. For other tryptophan residues including W21, W23 and W32, their replacement resulted in a greater loss in LnqQ's antibacterial activity, when compared to those of AucA (Fig. 2 and Supplementary Table S1). A plausible explanation is that the LnqQ is more sensitive to residue substitutions because of its monomeric nature, and so for future work the biophysical properties of LnqQ will be investigated.

## Conclusions

Analysis through racemic protein crystallography and antimicrobial susceptibility testing has allowed us to pinpoint the roles of each surface-exposed tryptophan residue in the model bacteriocin AucA. The importance of H-bond donation by tryptophan may be ubiquitous throughout the class of leaderless bacteriocins. The involving residues, particularly W40 and Lys44, are highly conserved, with presence in lacticin Q[25], lacticin Z[40] and epidermicin NI01[41] (Supplementary Table S2). Indeed, it may be a common mechanism used to interact with phospholipid head groups, as this residue is frequently found on the surface of other membrane-active antibacterial peptides[42–46], protein channels[8,47] and protein receptors[48–52].

Moreover, surface-exposed tryptophan can govern physiological roles other than membrane association. The non-conserved W3, whilst found to associate with tetrahedral ions, can also form part of the dimeric interface by interacting with residues of an adjacent molecule. This suggests that AucA is dimeric in solution and dissociates for phosphate head group binding in the presence of lipid membrane. For future work, the role of these key tryptophan residues will be investigated by use of model membranes composed of designated phospholipid, preparation of new site-directed variants and other relevant biophysical analysis. Finally, in complementation to recent studies[3,4], this work illustrates racemic protein crystallography is an effective approach to elucidate tryptophan interactions of membrane-associating proteins.

## Methods

**Preparation of bacteriocins and variants.** The preparation of bacteriocins by native chemical ligation was achieved by splitting the protein sequences into two peptide segments[31]. The N-terminal fragments (Met1-Ile10 for AucA, Met1-Trp23 for LnqQ) were prepared as a peptide hydrazide[53], and the C-terminal fragments (Cys11-Leu51 for AucA, Cys24-Lys53 for LnqQ) carried an Ala→Cys substitution to facilitate the ligation (see Supplementary Information for details of their synthesis). Briefly, in situ activation of the peptide hydrazide (2 µmol in 0.4 mL of

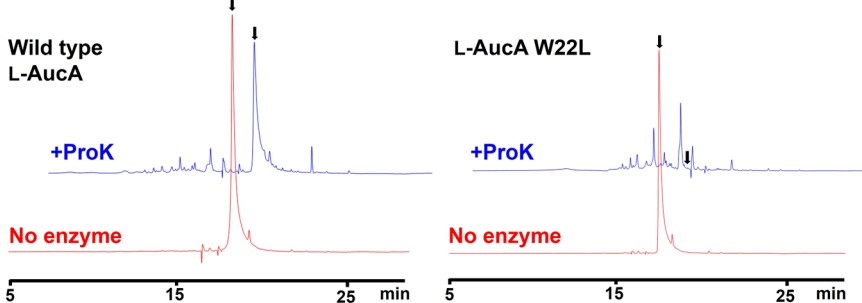

**Fig. 6 Comparing the proteolytic stability of the wild-type L-AucA and variant AucA-W22L.** HPLC chromatograms (UV 210 nm) of the bacteriocins are shown, following their incubation at 37 °C for 24 h with proteinase K (ProK) (blue), or the negative control that did not contain the enzyme (red). Peaks corresponding to intact bacteriocin are indicated by a black arrow.

6 M Gdn·HCl, 0.2 M NaPi, pH 3.0) was conducted by addition of 10 equiv. NaNO₂ at −15 °C for 20 mins. Thiol (4% v/v trifluoroethanthiol[54] or 2% v/v thiophenol, Supplementary Table S4) was added, the pH was adjusted to 5.0 and thioester formation allowed to proceed at room temperature for 10 mins. The C-terminal fragment (2 µmol in 0.4 mL of 6 M Gdn·HCl, 0.2 M NaPi, pH 7.0) was then added, the pH adjusted to 6.9 and ligation allowed to proceed at 37 °C for 4 h. Excess thiol was then removed (evaporation[54] or ether extraction, respectively) and the mixture degassed with Ar for 10 mins. Free-radical desulfurization was carried out by addition of 0.8 mL desulfurization buffer (0.2 M phosphate, 6 M Gdn·HCl, 0.4 M TCEP, 40 mM glutathione)[32]. The reaction was initiated by addition of VA-044 (80 mM) and tBuSH (80 mM) and heating on a shaker at 37 °C. The desulfurization was allowed to proceed for 16 h and complete conversion of cysteine to alanine was confirmed by LCMS. The reaction mixture was diluted 10-fold with water, passed through a 0.22 µM nylon filter and purified by semi-preparative HPLC. (Final isolated yields: AucA variants, 45–70%, 5.3–8.3 mg; LnqQ variants, 55–67%, 6.6–8.0 mg).

For preparation of full length AucA by automated SPPS (AucA-W22L, -W42L, -W3¹ᴹᵉW, -W31¹ᴹᵉW and -W40¹ᴹᵉW), the polypeptides were assembled using a Liberty Blue microwave peptide synthesizer (CEM corp.) on low loading (0.19 mmol/g) rink amide protide resin (CEM corp.). Ten equiv. of amino acid were used per coupling with DIC (20 equiv.) and Oxyma (10 equiv.) in DMF, and Fmoc deprotection conducted with 20% piperidine in DMF. Coupling was repeated for Ile, Val and all residues following 25 residues. For the coupling of Fmoc-Trp(Me)-OH (BACHEM, 4052226), the automated synthesis was paused, the resin removed from the vessel and the amino acid coupled manually (2 equiv. amino acid, 1.95 equiv, HATU and 4 equiv. DIPEA in DMF). The resin was then returned to the vessel and automated synthesis resumed (see Supplementary Methods for detailed synthetic procedures).

Synthesis of all bacteriocins and their variants were confirmed by LCMS, UPLC/ HDMS (Supplementary Data 6) and CD spectroscopy (Supplementary Information Figs. S2, S4 and S9).

**Racemic protein crystallography.** L-AucA and D-AucA were dissolved in water to a final concentration of 80 mg/mL. The peptide solutions were mixed 1:1 to yield an 80 mg/mL racemate of D- and L-AucA which was diluted two-fold with water to yield 40 mg/mL DL-AucA. Both 80 mg/mL and 40 mg/mL racemate concentrations were subject to sparse-matrix crystallization screening using Crystal Screen HT (HR2-130) and SaltRx HT (HR2-136) from Hampton research. 50 µL of each precipitant condition solution was added into the wells of a SWISSCI 96-well plate. The two racemate concentrations were each mixed 1:1 with the precipitant in a 0.4 µL sitting drop, yielding 384 crystallization drops across two screens. The best conditions which produced single, three-dimensional crystals were selected for optimization to produce crystals suitable for X-ray diffraction. LnqQ was crystallized in the same manner, using 27 mg/mL and 13.5 mg/mL of racemate, respectively. Details of crystallization conditions, X-ray diffraction data collection, and structure solution and refinement can be found in the Supplementary Methods. Data refinement statistics are given in the Supplementary Information Table S5 and the refined models of racemic AucA and LnqQ have been deposited in the Protein Data Bank with the PDB codes 8AVR, 8AVS, 8AVU, 8AVT and 7P5R (Supplementary Data 1–5).

**Bacteriocin minimum inhibitory concentration (MIC) assay.** Minimum Inhibitory Concentrations (MIC) were determined for each of the bacteriocins and controls using a broth microdilution method, as per CSLI guidelines for microbial susceptibility testing[55]. The peptides were dissolved in sterile deionized water to prepare the working stocks, with concentrations determined by UV absorbance at 280 nm[56]. Overnight cultures from the strains of interest were incubated with shaking overnight at 37 °C. Cation-adjusted Mueller Hinton Broth 2 (MHB 2) was used for the S. aureus, and S. epidermidis strains and Brain Heart Infusion (BHI) broth was utilized for the E. faecalis and E. faecium strains. The cultures were prepared and diluted to the turbidity of a 0.5 McFarland standard using MHB or BHI fresh media. Increasing concentrations of the leaderless bacteriocins were added to a 96-well polypropylene plate. The range tested for each bacteriocin was 64–0.125 µg/mL. The bacterial culture was further diluted and mixed in a 1:1 ratio with the bacteriocin solution to yield a final concentration in the wells of $5 \times 10^5$ CFU/mL. Ampicillin, tetracycline and melittin were used as positive controls and included in each assay. A 1:1000 dilution of the growth control was prepared in the respective media and plated onto agar (100 µL) to ensure the correct inoculum concentration ($5 \times 10^5$ CFU/mL indicated by ≈ 50 bacterial colonies). The 96-well and agar plates were incubated at 37 °C, and growth was assessed by the formation of a pellet observable with the naked eye after 20 h. Uninhibited bacterial growth at 64 µg/mL was denoted as MIC > 64 µg/mL and for the purposes of this work was classified as loss of activity. Each assay was performed in triplicate. The average MICs (µg/mL) against each of the tested strains are reported in Fig. 2 and Supplementary Information Table S1.

For full detailed experimental procedures and analytical data, see supplementary information file.

**Reporting summary**. Further information on research design is available in the Nature Portfolio Reporting Summary linked to this article.

## Data availability

Supplementary Figures and Methods are available in the Supplementary Information file. Model coordinates and structure factors for the racemic protein crystal structures reported in this work have been deposited in the Protein Data Bank as entries 8AVR (Supplementary Data 1), 8AVS (Supplementary Data 2), 8AVT (Supplementary Data 3), 8AVU (Supplementary Data 4) and 7P5R (Supplementary Data 5). Peptide and protein LCMS data are available in Supplementary Data 6.

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

## Acknowledgements

The authors thank Diamond Light Source for access to beamlines iO3 and iO4-1 (Proposal Number: mx20147), and Dr. Pierre Rizkallah and Dr. Patrick Baumann for their assistance with X-ray diffraction data collection. The authors also thank Prof. G. Michael Blackburn for comments on the manuscript. A.J.L. is funded by an EPSRC scholarship (2107414). L.D.M. and B.L.F. thank the Québec Fonds de Recherche—Nature et Technologies (319170/B1X) and Santé (269182) for financial support, respectively. Y.J. thanks the Wellcome Trust (218568/Z/19/Z) for the financial support. L.Y.P.L. thanks BBSRC (BB/T015799/1) for the financial support.

## Author contributions

A.J.L. performed synthesis and physical characterization of the bacteriocins. X.L. conducted synthesis of the wild-type and enantiomeric LnqQ. A.J.L. and Y.J conducted the racemic protein crystallography. L.D.M. conducted the bacteriocin MIC assays. I.M.T. conducted the protease digestion experiments. B.L.F., Y.J. and L.Y.P.L. supervised the studies. A.J.L., Y.J. and L.Y.P.L. wrote the manuscript. All authors contributed to editing and critical proofreading of the manuscript.

## Competing interests

The authors declare no competing interests.
