## [Peer Review File · Communications Chemistry]

Reviewers' comments:

Reviewer #1 (Remarks to the Author):

In this manuscript, racemic protein crystallography and antimicrobial susceptibility testing were used to determine the structure-activity relationship of each tryptophan residue in the model bacteriocin protein, AucA. AucA exhibits broad-spectrum activity against Gram-positive pathogens and contains conserved tryptophan residues (W22, W31, and W40). This work shows that H-bond donation of W40 is essential for antibacterial activity and W3 is essential for AucA oligomerization and proteolytic resistance by forming an intermolecular H-bond network. I recommend the following minor changes before publication.

1. In the second paper, in the case of *S. aureus* CW115852, D-AucA displayed an eight-fold increase in activity over that of its native L-enantiomer. The authors speculate that this is because the D-counterparts are more resistant to degradation. It would be nice to have additional experiments to support this hypothesis.
2. In the third paragraph on page 2, the authors mention that "The highly conserved W31 and W44 (Table S2) are.....". However, in Table S2, the conserved tryptophan residues are W22, W31, and W40.
3. The authors mention that "when AucA crystals were grown in conditions using citrate or acetate, the equivalent network was not observed". The structure of AucA grown in citrate or acetate condition is needed to support this important claim.
4. The legend of Figure 1 shows "e) dimer AucA (PDB: 8AVU)", but the dimeric AucA, e, is missing from Figure 1...
5. The correspondence of the following figures is inconsistent. Figure S5, S6, S7, S8, Table S4 and "The AucA-W22L variants is considerably morehydrolysis (Figure 3 and S8)".
6. The H-bond of W3 in Figure S5 is missing.
7. The colors used in Figure S7 are inconsistent.
8. There are duplicate acknowledgment sections.

Reviewer #2 (Remarks to the Author):

The manuscript submitted by Lander et al. investigates the role of five Trp residues in the sequence of the antimicrobial proteins aureocin A53 and related lactacin Q. Their study relies essentially on racemic protein crystallography and antimicrobial tests. Crystal structures reveal Trp residues can engage in several types of non-covalent interactions with positively charged side chains of lysines in the protein sequences and tetrahedral ions introduced as mimics of phospholipid polar headgroups. The authors relate these interactions to the antimicrobial activity using different mutants of AucA where Trp is substituted with other natural amino-acids or unnatural analogues. Overall, this study highlights very interesting points on the multiple roles Trp can play in membrane interactions and related antimicrobial activity but some interpretations are very speculative and not sufficiently supported by experiments. The results could also be discussed deeper and put more in perspective with the literature on related bacteriocins and the role of Trp in membrane active peptides and proteins. More specifically, here are some points that should be addressed before the manuscript is suitable for

publication in Communications Chemistry.

1/ Although the synthesis involving NCI is well described in the Methods section, a Figure summarising the strategy in the Results and Discussion section would be welcome.

2/ The structure of AucA in the presence of tetrahedral anions revealed interesting tripartite interactions involving Trp and Lys residues and the anions. The author claims these anions are structural analogues of the headgroups of phospholipid, I do not agree. They carry two negative charges, which is not the case of the headgroup of PG for example, which is representative of Gram+ membranes. The observed interactions are very interesting but it is a shortcut to directly suggest such H-bonding networks also occur with phospholipids and have a physiological role. The authors should be less speculative on these aspects.

3/ Crystallography data suggest Trp is involved in H-bonding networks, but the authors do not discuss π -type interactions involving the aromatic ring. The authors should discuss possible π interactions with side chains of Lys and/or sulfate/phosphate anions.

4/ A lot of assumptions about lipid interactions are based on antimicrobial activity. It would be more convincing if the authors also studied the interactions of the different mutants with model membranes of controlled lipid composition. There are many experimental approaches that could be easily implemented to complete this aspect.

5/ Substituting Trp with Glu impacts the isoelectric point of the protein, not surprisingly, it abolishes the antimicrobial activity of the protein. Substituting W40 with N-Me-Trp reduces the antimicrobial activity but does not completely abolish it, the aromatic ring could also play a role (see point 3). What would be the effect of a substitution with a non-aromatic H-bond donor like Gln?

6/ The authors suggest W3 plays a role in protein oligomerisation, which is an interesting point but raises several questions. The dimer-to-monomer transition could occur in a membrane environment, it would be good to have direct experimental evidence of this.

Did the authors consider designing mutants with increased dimer-state stability that could be assessed by SEC? Finally, the results presented here do not agree with the structural data obtained by NMR in 2016, this should be discussed more in depth.

7/ When discussing protein stability regarding proteolysis, it would be more convincing to assess the stability of the mutants in culture supernatants.

Minor points:

1/ Please provide the sequence of AucA and LnqQ in the main text

2/ The legend in Figure 1 refers to a panel e) which is not displayed on the figure

3/ To summarise the antimicrobial activity data in a single value, the authors could use the geometric mean of the MICs presented in table 1

4/ Page numbers would be helpful

Reviewer #3 (Remarks to the Author):

The paper 'Roles of inter- and intramolecular tryptophan interactions in membrane-active proteins revealed by racemic protein crystallography' by Louis YP Luk et al. uses a combination of chemical protein synthesis and racemic crystallography to elucidate original roles of conserved surface tryptophan residues on two bacteriocin molecules that interact with membrane phospholipids.

The methods used to introduce unnatural amino acids are original and results are novel. The data support the author's conclusions.

The paper provides convincing evidence for its conclusions, which will influence thinking about how bacteriocins work.

The manuscript is important to scientists in that specific sub-field of chemistry and are of significance for the continuing development of novel antibiotics.

Queries:

MAIN TEXT

Why was racemic crystallography used? Explain.

Why was the stability of D-AucA to proteases never examined?

METHODS

The first sentence should read as follows: 'The preparation of bacteriocins by native chemical ligation was achieved by condensing two unprotected synthetic peptide segments to form the full-length polypeptide chains.'

i.e. Use 'segments' not 'fragments'.

SUPPLEMENTARY INFORMATION

1. How was pH measured?

If by a glass electrode, in 6 M guanidine.HCl the true H⁺ ion concentration is that corresponding to ~0.8 pH units higher than the reading from the glass electrode. If you adjusted to pH 7.0 based on the glass electrode reading, the true reaction pH was 7.8 [see; "pH Corrections in Chemical Denaturant Solutions" Acevedo, et al., Analytical Biochemistry 306, 158–161 (2002)]

2. Why do the enantiomers L-aureocin A53 (19.47 min Fig. S13.2) and D-aureocin A53 (20.57 min Fig. S14.2) have differing UPLC elution times?

What happens if you co-inject the enantiomers for UPLC?

Annotated pdf files for the Main Text and Supplementary Information, with suggested corrections to improve clarity and other minor comments, are attached.

Response to reviewer comments on article titled: “Roles of inter- and intramolecular tryptophan interactions in membrane-active proteins revealed by racemic protein crystallography”

Dear reviewers,

We would like to thank the valuable feedback provided the reviewers. In particular, to address the major comment(s), we have obtained experimental data that clearly demonstrated the susceptibility of L-AucA to proteolysis, while the enantiomeric D-counterpart remained intact within the examined timeframe (24 hours). A point-by-point response to each of the reviewers' comments are detailed below:

Reviewer 1 recommends the following minor changes before publication.

1. In the second paper, in the case of *S. aureus* CW115852, D-AucA displayed an eight-fold increase in activity over that of its native L-enantiomer. The authors speculate that this is because the D-counterparts are more resistant to degradation. It would be nice to have additional experiments to support this hypothesis.

We have now tested the proteolytic stability of L-AucA and its D-counterparts. They are summarised as follows and on Pg 2 in the manuscript:

“The stability of L- and D-AucA was evaluated by subjecting them to different proteases (Figure S3). L-AucA was found to be proteolytically resistant when treated with proteinase K. Chymotrypsin only caused degradation of L-AucA when the enzyme:substrate ratio increased from 1:20 to 10:1. Similarly, trypsin degraded L-AucA at an enzyme:substrate ratio of 10:1. In contrast, the enantiomeric D-AucA remained intact under all conditions tested (Figure S3). These findings suggest that the unusual chirality of D-AucA confers an even more superior resistance to proteolysis compared to its relatively stable L-counterpart.”

We have also added the following sentences on Pg 5:

“Indeed, the monomeric L-LnqQ appeared to be more sensitive to proteolytic degradation, as its enantiomeric D-counterpart is resistant to proteolytic degradation and is significantly more potent leading up to 16-fold of difference in activity (Table 1 and S1).”

Also on Pg S2 in SI:

Figure S1: Bacteriocin proteolytic stability assays showing different susceptibility of the enantiomers. LC traces at 210 nm of protease reaction quenched after 5- and 24-hours incubation of bacteriocin with trypsin or chymotrypsin at 25 °C. HPLC conducted using a 5-70% gradient of A/B over 30 minutes on a RP-C4 column (ACE, 4.6 mm x 250 mm, 300 Å, 5 μm).

2. In the third paragraph on page 2, the authors mention that “The highly conserved W31 and W44 (Table S2) are.....”. However, in Table S2, the conserved tryptophan residues are W22, W31, and W40.

Thank you for pointing out this typographical error. We have now rephrased the manuscript as follows:

“The highly conserved W31 and W40 (Table S2) are hydrogen bonded (<3.2 Å) to the negatively charged sulfate ion along with the ammonium groups of neighboring lysine residues (K27, K44; Figure 1a-c).”

3. The authors mention that “when AucA crystals were grown in conditions using citrate or acetate, the equivalent network was not observed”. The structure of AucA grown in citrate or acetate condition is needed to support this important claim.

The structure of AucA grown in the following condition as described on Pg S14:

“0.2 M ammonium acetate, 0.2 M sodium citrate and 29% PEG 4000 v/v at pH 5.6,”

The PDB file has now been deposited into the protein data bank under the accession code **8AVS**. X-ray coordinate files and electron density maps are made freely available to the reader. Please see the ESI for further details.

4. The legend of Figure 1 shows “e) dimer AucA (PDB: 8AVU)”, but the dimeric AucA, e, is missing from Figure 1...

Thank you for spotting the typo. The caption in now Figure 2 (originally Figure 1) has been amended.

5. The correspondence of the following figures is inconsistent. Figure S5, S6, S7, S8, Table S4 and “The AucA-W22L variants is considerably morehydrolysis (Figure 3 and S8)”.

The referencing of figure captions have been revised.

6. The H-bond of W3 in Figure S5 is missing.

Figure S5 contained the interaction radar for the dimeric interface, whereas Figure S2 is the analysis of AucA dimeric interface (PDB: 8AVU) using Protein Interfaces, Surfaces and Assemblies (PISA) software¹ in the CCP4 software suite.

7. The colors used in Figure S7 are inconsistent.

Thank you for pointing out his issue. However, the Agilent software with which this figure was constructed does not provide the option to alter the LC trace colors. We aim to clarify the identity of these traces by colour-coordinated labels for clarity.

8. There are duplicate acknowledgment sections.

Thank you, the duplicate has been deleted.

Reviewer #2 indicated some points should be addressed before the manuscript is suitable for publication in *Communications Chemistry*.

1. Although the synthesis involving NCI is well described in the Methods section, a Figure summarising the strategy in the Results and Discussion section would be welcome.

A figure has been added as **Figure 1** in the results and discussion.

2. The structure of AucA in the presence of tetrahedral anions revealed interesting tripartite interactions involving Trp and Lys residues and the anions. The author claims these anions are structural analogues of the headgroups of phospholipid, I do not agree. They carry two negative charges, which is not the case of the headgroup of PG for example, which is representative of Gram+ membranes. The observed interactions are very interesting but it is a shortcut to directly suggest such H-bonding networks also occur with phospholipids and have a physiological role. The authors should be less speculative on these aspects.

Thank you for this thoughtful insight. Sulfate and glycerol 3-phosphate are isosteric to many phospholipid headgroups of Gram-positive bacteria. However, as the reviewer pointed out, they may not contain the charges as that of the phospholipid. Hence, we agree with the reviewer that the interactions between tryptophan and phospholipid head group are speculative, and hence the interactions were further investigated in this work by the residue substitution experiments. Indeed, replacement of the N-H donation with a methyl group in the L-¹MeW40 variant did substantially eliminate a majority of the AucA activity. Furthermore, we have now included the AucA crystal structure grown in the absence sulfate for reviewer's investigation (see point 3 above). Lastly, we have adopted a more tentative tone (**highlighted green**) in our revised manuscript including the text below:

"Since sulfate and glycerol 3-phosphate are isosteric to many of the phospholipid head groups in bacterial membranes, these observations suggest a physiological role of W3, W31 and W40 that interact with the membrane lipid through H-bond donation."

3. Crystallography data suggest Trp is involved in H-bonding networks, but the authors do not discuss π -type interactions involving the aromatic ring. The authors should discuss possible π interactions with side chains of Lys and/or sulfate/phosphate anions.

Based on the orientations of the indole and ammonium group, we did not observe π -type interactions with lysine ammonium groups or solute anions. The following sentence has been added to Pg 2:

"While there are no observable anion- π or cation- π interactions between tryptophan and the sulfate or lysine side chains, the K44 aliphatic side chain associated with W40 via van der Waals interactions, as also suggested in previous NMR studies,²⁷ which could aid correct orientation for cooperative interaction with anions."

4. A lot of assumptions about lipid interactions are based on antimicrobial activity. It would be more convincing if the authors also studied the interactions of the different mutants with model membranes of controlled lipid composition. There are many experimental approaches that could be easily implemented to complete this aspect.
5. Substituting Trp with Glu impacts the isoelectric point of the protein, not surprisingly, it abolishes the antimicrobial activity of the protein. Substituting W40 with N-Me-Trp reduces the antimicrobial activity but does not completely abolish it, the aromatic ring could also play a role (see point 3). What would be the effect of a substitution with a non-aromatic H-bond donor like Gln?

6. The authors suggest W3 plays a role in protein oligomerisation, which is an interesting point but raises several questions. The dimer-to-monomer transition could occur in a membrane environment, it would be good to have direct experimental evidence of this. Did the authors consider designing mutants with increased dimer-state stability that could be assessed by SEC?

We appreciate the reviewer's suggestion. However, this article focuses specifically on the application of racemic protein crystallography to investigate tryptophan interactions. Exploring the potential of these suggested methods is an intriguing avenue to consider for our future studies. We have added the following statement to Pg 5:

"For future work, the role of these key tryptophan residues will be investigated by use of model membranes composed of designated phospholipid, preparation of new site-directed variants and other relevant biophysical analysis."

7. Furthermore, the results presented here do not agree with the structural data obtained by NMR in 2016, and we have added more in-depth discussion to draw readers' attention.

Many thanks for raising this point. **This is now highlighted on Pg 2 & 4.**

8. When discussing protein stability regarding proteolysis, it would be more convincing to assess the stability of the mutants in culture supernatants.

Based on our analysis of the culture supernatant, we found that the bacteriocins remained stable. This may be due to a number of factors: (1) deactivation of the bacterial enzymes during preparation of supernatants for proteolytic assay, (2) the high bacteriocin concentrations used for the assays (1.0 mg/mL, 10x the injected concentration to dilute the cultures for HPLC analysis) which do not accurately represent the real enzyme to substrate ratios (<64 µg/mL used for MIC), (3) the bacterial enzymes may not be secreted into the media (e.g membrane proteases). Hence, we replaced this study with the treatment of high concentrations of proteases, including trypsin and chymotrypsin, which clearly demonstrated the superior stability of the D-counterparts. Please see **Pg 2** in the manuscript for further discussion.

Minor points:

1. Please provide the sequence of AucA and LnqQ in the main text

The sequence of AucA is now added in Figure 1, alongside a synthetic scheme overview. Since LnqQ is not the focus of this manuscript, its sequence remains in Table S2 in the SI.

2. The legend in Figure 1 refers to a panel e) which is not displayed on the figure

Thank you for highlighting this typographical error. This has since been **resolved**.

3. To summarise the antimicrobial activity data in a single value, the authors could use the geometric mean of the MICs presented in Table 1

Thank you for the interesting idea. However, for the purposes of this study, many MIC are denoted as >64µg/mL, classified as substantial loss of activity (inactive in the range tested). This would obscure any calculated geometric mean across multiple bacterial strains.

4. Page numbers would be helpful

Page numbers have been added.

Reviewer #3 (Remarks to the Author) approved for publication and has the following queries:

1. Why was racemic crystallography used? Explain.

There are two main reasons for using racemic protein crystals:

- (1) Ease of protein crystal formation. Racemic protein crystals are known for their ease of formation, which allows us to investigate specific intra- and intermolecular interactions.
- (2) High resolution structure. The tight packing of enantiomeric protein molecules within the crystal lattice provides an ideal environment for revealing intricate molecular interactions.

The discussion was included on Pg 1 and the references therein.

2. Why was the stability of D-AucA to proteases never examined?

This is now included on Pg 2.

METHODS

3. The first sentence should read as follows: 'The preparation of bacteriocins by native chemical ligation was achieved by condensing two unprotected synthetic peptide segments to form the full-length polypeptide chains.'

i.e. Use 'segments' not 'fragments'.

This has been resolved, see Pg 2, 5 and 9.

SUPPLEMENTARY INFORMATION

1. How was pH measured?

If by a glass electrode, in 6 M guanidine.HCl the true H⁺ ion concentration is that corresponding to ~0.8 pH units higher than the reading from the glass electrode. If you adjusted to pH 7.0 based on the glass electrode reading, the true reaction pH was 7.8 [see; "pH Corrections in Chemical Denaturant Solutions" Acevedo, et al., Analytical Biochemistry 306, 158–161 (2002)]

Thank you for highlighting this. Indeed, we assume a value of approximately 0.8 units lower than that measured by glass electrode. The following has been added to the "materials and instruments" of the ESI on Pg 11:

The pH measurements were conducted using a Mettler Toledo FiveEasy Plus pH meter fitted with an FP20-Micro glass electrode. For measurements in 6 M Gdn-HCl, the measured pH value was assumed to be 0.8 units lower than the actual value.

2. Why do the enantiomers L-aureocin A53 (19.47 min Fig. S13.2) and D-aureocin A53 (20.57 min Fig. S14.2) have differing UPLC elution times?
What happens if you co-inject the enantiomers for UPLC?

The LC traces of the L and D proteins were collected with a time gap of several months, which suggests a change in LC quality. In general, when subjected to LC analysis, the L- and D-proteins are expected to elute at similar or overlapping retention times.

I hope these amendments have made the article suitable for acceptance. If you require any additional information, please do not hesitate to contact us.

Yours sincerely,

Louis Luk, PhD, on behalf of the authors

REVIEWERS' COMMENTS:

Reviewer #1 (Remarks to the Author):

The authors have addressed all my concerns.

Reviewer #2 (Remarks to the Author):

I am happy with the latest version of the manuscript which should be accepted for publication.

Reviewer #3 (Remarks to the Author):

The responses of the authors and revisions to the manuscript are satisfactory.